# Selective monodeuteration enabled by bisphosphonium catalyzed ring opening processes

Yuanli Xu[1,4], Wenlong Chen[2,4], Ruihua Pu[3], Jia Ding [2], Qing An[2], Yi Yang [1] ✉, Weimin Liu [3] ✉ & Zhiwei Zuo [2] ✉

The selective incorporation of a deuterium atom into small molecules with high selectivity is highly valuable for medical and chemical research. Unfortunately, this remains challenging due to the complete deuteration caused by commonly used hydrogen isotope exchange strategies. We report the development of a photocatalytic selective monodeuteration protocol utilizing C–C bond as the unconventional functional handle. The synergistic combination of radical-mediated C–C bond scission and deuterium atom transfer processes enables the effective constructions of benzylic CDH moieties with high selectivity for monodeuteration. The combinational use of a bisphosphonium photocatalyst, thiol catalyst, and $CH_3OD$ deuteration agent provides operationally simple conditions for photocatalytic monodeuteration. Moreover, the photoinduced electron transfer process of the bisphosphonium photocatalyst is elucidated through a series of spectroscopy experiments, identifying a peculiar back electron transfer process that can be regulated by subsequent nucleophilic additions.

The selective incorporation of deuterium atoms into small molecules is of great importance in medicinal and chemical research[1–4]. Compared to their nondeuterated analogues, deuterium-labeled compounds exhibit unique metabolism and pharmacokinetic properties, which have been frequently exploited to increase the bioavailability of pharmaceutical candidates[5,6]. Deuterated compounds have also been used as metabolic tracers and mass spectrometry analytic standards[7]. In organic chemistry, kinetic isotope effect (KIE) experiments with mono-deuterated or fully-deuterated compounds have been widely used in mechanistic investigations to elucidate reaction pathways[8,9]. Despite widespread uses, the preparation of deuterium-labeled molecules with high selectivity oftentimes are rather challenging tasks, as the separation of unlabeled or partially labeled materials from the desired deuterated product has been impeded by the trivial difference in physical properties.

With the advancement of transition metal catalysis, tremendous progress has been made in H/D exchange reactions, enabling various environmentally benign and operationally simple protocols[2,10–18]. In this manifold, the selectivity problem can be addressed through the complete deuteration of an aliphatic carbon which can be easily achieved by thermodynamic effect[19], while the selective incorporation of a single deuterium into methylene and methyl groups is deemed impractical due to over-deuteration (Fig. 1)[20]. Thusly, the preparation of monodeuterated molecules heavily relies on the use of pre-functionalized substrates such as alkyl halides. The necessities of expensive deuteration reagents as well as multistep conversions has raised urgent needs to develop operationally simple and sustainable methods[21–30]. Recently, selective monodeuteration of $C(sp^3)$–H bonds have been realized through elegantly designed intramolecular 1,5-HAT processes by Studer, Xie, and Zhu groups[31,32]. With the combinational

[1]Innovation Center for Chenguang High Performance Fluorine Material, Key Laboratory of Green Chemistry of Sichuan Institutes of Higher Education, Sichuan University of Science and Engineering, 643000 Zigong, China. [2]State Key Laboratory of Organometallic Chemistry, Shanghai Institute of Organic Chemistry, Chinese Academy of Sciences, 200032 Shanghai, China. [3]School of Physical Science and Technology, ShanghaiTech University, 201210 Shanghai, China. [4]These authors contributed equally: Yuanli Xu, Wenlong Chen ✉e-mail: yangyiyoung@163.com; liuwm@shanghaitech.edu.cn; zuozhw@sioc.ac.cn

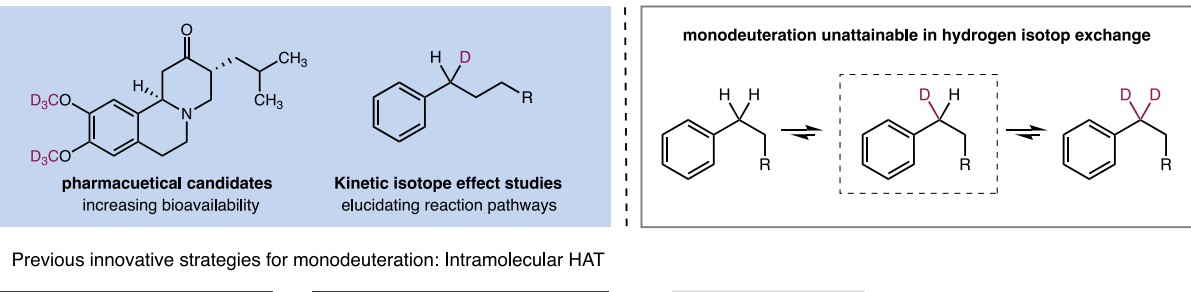

a. Selective deuteration: an empowering tool in medicinal and chemical research and a challenging task in organic synthesis

**pharmaceutical candidates**
increasing bioavailability

**Kinetic isotope effect studies**
elucidating reaction pathways

monodeuteration unattainable in hydrogen isotop exchange

b. Previous innovative strategies for monodeuteration: Intramolecular HAT

Studer

AIBN, ArSH, Δ

Xie, Zhu

Ir photocatalyst, RSSR, LED

1,5-HAT

c. Photocatalytic ring opening protocol for selective monodeuteration

photocatalyst, HAT catalyst

deuteration reagent: CH₃OD

**aliphatic rings**
easily accessed materials

**mono-deuterated motif**
distally functionalized building block

- high selectivity for monodeuteration
- 3- to 7-membered cyclic substrates
- distal functional group for futher manipulation
- operationally simple conditions

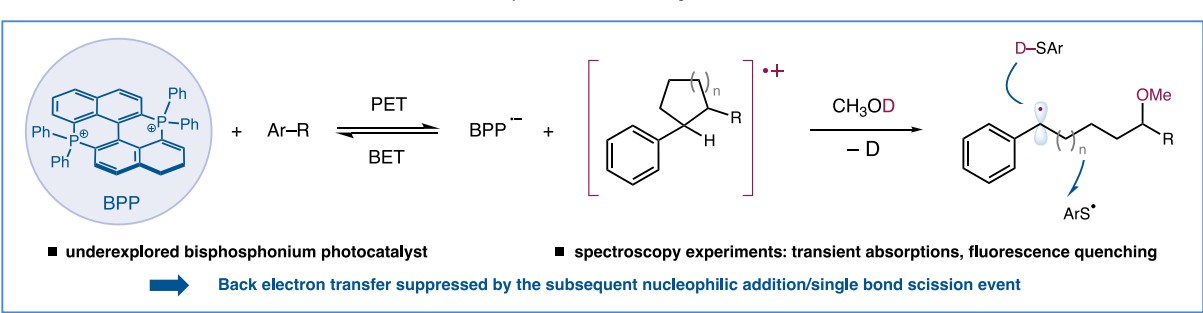

BPP

PET / BET

BPP

CH₃OD / − D

D−SAr

ArS•

- underexplored bisphosphonium photocatalyst
- spectroscopy experiments: transient absorptions, fluorescence quenching

**Back electron transfer suppressed by the subsequent nucleophilic addition/single bond scission event**

**Fig. 1 | Selective monodeuteration via photoredox catalysis. a** Selective deuteration: an empowering tool in medicinal and chemical research and a challenging task in organic synthesis. **b** Previous innovative strategies for monodeuteration: Intramolecular HAT. **c** Photocatalytic ring opening protocol for selective monodeuteration.

uses of radical precursors with AIBN, or amide substrates with Ir photocatalyst, monodeuterated amides can be obtained with high selectivity. Considering the peculiar uses of mono-deuterated compounds in KIE studies and pharmaceutical investigations[5,9], the development of innovative strategies and catalytic protocols remains in high demand.

Radical-mediated C−C bond cleavage and functionalizations, with emerging photoredox catalysis as the enabling platform to generate open-shell radical intermediates, have provided a synthetically valuable strategy to exploit ubiquitous C(sp³)−C(sp³) bonds as functional handles[33,34]. Under ambient temperature, a high-energy radical species such as alkoxy radical or benzene radical cation could promote the homolytic cleavage of the adjacent C−C single bond to generate a more stabilized carbon-centered radical[35,36]. Importantly, the resultant alkyl radical intermediate could undergo diversified conversions which have provided intriguing opportunities for functional group installations. As such, a series of ring-opening transformations have been developed over the last decade, demonstrating various intriguing applications, including oxidations, halogenation, alkylation, and arylation for the syntheses of distally functionalized carbonyl compounds[37−56]. We recently wondered whether a radical-mediated deuterium transfer event could be incorporated in the photocatalytic ring opening process for the

selective mondeuteration, providing an intriguing strategy to utilize inert and ubiquitous C(sp³)−C(sp³) bonds as functional handles for deuterium incorporation. Compared to radical-mediated H/D exchange where conversion rates suffer from the small driving force (difference in zero-point energy, ~1 kcal/mol), ring-opening deuterations take advantage of the substantial energy gain in strain release (strain energies of 3- to 5- membered aliphatic rings, 6-27 kcal/mol) to enhance efficient deuteration.

Given the prevalence of benzylic C−H bonds in pharmaceutical candidates, we opt to first investigate the ring-opening deuteration protocol for the selective constructions of benzylic CDH moieties. In the deuteration of aromatic hydrocarbons, the relatively weakened benzylic C−H bonds can facilitate H/D exchange but have posed severe challenges for the selective monodeuteration[2]. We envisioned that the well-established β-bond cleavage of aromatic radical cations could render a facile approach to achieve benzylic monodeuteration, via selective ring openings of cyclic hydroaromatics and the intermediacy of a benzylic radical intermediate[35,57−62]. Through the use of efficient HAT catalysts, inexpensive and operationally handy deuteration agents such as CH₃OD can be employed to enforce the desired deuterium atom transfer. Critically, the selective and irreversible C−C bond scission process would only allow the installation of one deuterium atom. Regarding the challenges raised by the high oxidation

potential of aromatic hydrocarbons, we firstly looked into the development of highly oxidizing photocatalysts.

Organophosphorus compounds have been extensively investigated in organic electronics, optical sensing, and imaging applications[63–68]. Nevertheless, their unique photophysical and electrochemical properties remain largely underexploited in photoredox catalysis. Recently, we found that a bisphosphonium (BPP) compound easily prepared from the oxidative cyclization of 2,2′-*bis*(diphenylphosphino)-1,1′-binaphthyl (BINAP), can act as a strongly oxidizing organophotocatalyst to promote the intramolecular hydroetherification of alkenols. The remarkable capacity of this photoredox catalyst, including high oxidation potential ($E^*_{1/2} = 2.17$ V vs. SCE), strong absorption of visible light ($\lambda_{max} = 413$ nm), and long excited life time, has not been fully explored for synthetic transformations[69–72].

Herein, we disclose a practical and selective monodeuteration protocol by bisphosphonium photocatalysis which can enable the facile constructions of benzylic CDH moieties from easily accessed materials. The photoexcitation and photoinduced electron transfer events of bisphosphonium catalyst are elucidated in its premiere instance by spectroscopy experiments, paving the way for future applications.

## Results and discussion
### Reaction optimization
To validate our hypothesis, phenylcyclopropane ($E_{1/2} = 1.7$ V vs. SCE) was employed as the template substrate, readily available monodeuteromethanol $CH_3OD$ was chosen as the deuteration reagent. In combination with commonly used atom transfer catalyst (TRIPS)$_2$, a variety of oxidizing photocatalysts were evaluated under the irradiation of high intensive LED light (see supporting information for the detailed description of set-up for the parallel reactions). As depicted in Fig. 2, commonly utilized oxidizing photocatalysts Ru(bpz)$_3$$^{2+}$ and triphenylpryrlium were found ineffective, while mesitylacridinium photocatalyst could afford product **2** with 68% yield and 95% D-incorporation. Among the P-containing conjugated arenes evaluated, BPP photocatalyst rendered the optimal condition, delivering the monodeuterated product with high efficiency and selectivity (83% yield, 96% D-incorporation). A seemingly positive correlation between the oxidation capacity and catalytic efficiency can be found in this series of organophotocatalyst, as phosphonium salt **3** ($E^*_{1/2} = 1.54$ V vs. SCE) was found inactive and bisphosphapyrenium **4** ($E^*_{1/2} = 1.69$ V vs. SCE) enabled moderate efficiency (73% yield, 96% D-incorporation). Furthermore, control experiments have indicated the essential role of BPP photocatalyst, thiol catalyst and LED light. Notably, the control experiment with $H_2O$ in entry 4 has also revealed the critical importance of anhydrous conditions to achieve high deuteration selectivity.

### Selective monodeuteration of cyclic aromatic substrates
With the optimal condition in hand, we next explored the scope of cyclic aromatic substrates and were delighted to find that this operationally simple and inexpensive catalytic system could be adapted to 3-membered to 7-membered cyclic starting materials. Importantly, a high degree of monodeuteration selectivity was achieved across the board, delivering the desired products with more than 95% D-incorporation. Even in the presence of weakened benzylic C−H bonds and acidic α-carbonyl C−H bonds which are prone to undergo H/D exchange to induce multiple deuterium incorporation, only the desired monodeuterated product was obtained. As shown in Fig. 3, this organophotocatalytic monodeuteration protocol can be successfully applied to a variety of substituted arylcyclopropanes, generating monodeuterated 3-methoxypropylbenzenes in high D-incorporation rate. Arylbornic ester (**7, 8, 14**) and aryl bromide (**9, 10, 26**), commonly used functional handles in transition metal catalysis, could be well tolerated under the current photocatalytic condition. Owing to the high oxidation capacity of BPP catalyst, electron deficient arenes which

are more challenging to activate in SET oxidations can be employed, although $p$-CF$_3$ and $m$-Br substitutes rendered somewhat declined efficiencies. Regarding unsymmetrical arylcyclobutanones, the radical cation-mediated ring-opening process proved to be highly selective, as the C−C bond between the carbonyl and benzylic carbon terminus were selectively cleaved to generate monodeuterated 4-phenylbutanoates (**16-27**). For less strained 5-, 6-, 7-membered cyclic substrates, we noticed that this photocatalytic ring-opening deuteration can be effectively promoted by stabilizing the partially formed positive charge at the β-carbon terminus[73]. The introduction of a methoxy group at the β-carbon led to the facile cleavage of these less strained C−C bonds, rendering monodeuterated products with a distal hemiacetal functionality for further functionality manipulations (see Figs. S19–S23 for the conversions into amine, alcohol, alkene and tetrahydroquinoline moites). A variety of N-heterocyclic compounds, including pyridine and quinoline derivatives, have been effectively incorporated into monodeuterated products with yields ranging from excellent to moderate, without a significant impact on deuterium incorporation efficiency (**33-36**). Deuteration using $D_2O$ as the agent under standard conditions has yielded satisfactory outcomes (**37, 38**). Furthermore, $^{18}O$ labeling is readily accomplished with $H_2^{18}O$ as the nucleophile under standard conditions (**39**). Importantly, the cyclic hydroaromatic moieties embedded in complex scaffolds such as steroid ring systems can be selectively cleaved, enabling the selective incorporation of benzylic CHD moieties. The implementation of a continuous-flow synthesis method has facilitated scaled-up production of **2**, achieving an 87% yield and a space-time yield of 17.4 g/L•h under optimized conditions, while significantly reducing the disulfide loading to 5 mol%.

### Spectroscopy experiments
Spectroscopy experiments were then conducted to probe the critical photoinduced electron transfer (PET) event between phenylcyclopropane **1** and BPP catalyst. Unexpectedly, Stern–Volmer quenching studies revealed an insignificant fluorescence quenching effect of phenylcyclopropane even at relatively high concentrations (Fig. 4A), reminding us that the electron transfer between excited BPP and phenylcyclopropane might be sluggish. This was in striking contrast to the observation made in the same set of experiments with triethylamine as the reductant (see Fig. S4 for the quenching experiment with triethylamine). As demonstrated in Fig. 4B, the reduced BPP ([BPP]$^{•−}$) generated from the PET process with triethylamine can be clearly identified in the 580 nm region in the UV-vis absorption spectrum, while the irradiation of the solution containing BPP and phenylcyclopropane did not cause any observable changes in the absorption spectrum. From the redox potential perspective, both PET processes are thermodynamically favorable, but only the PET process with triethylamine led to the net generation of [BPP]$^{•−}$. Considering that the only difference lies in the oxidative nature of the resultant radical cations, we posited that highly oxidizing aromatic radical cations might result in an overwhelming back electron transfer, leading to declined efficiency[74,75]. These preliminary findings, contradicting the efficient ring-opening deuteration we obtained, promoted us to elucidate the PET events between BPP photocatalyst and the phenylcyclopropane substrate.

As the excitation pattern of bisphosphonium photocatalyst is currently unknown, we decided to first probe the photoexcitation process via ultrafast transient absorption (TA) spectroscopy. As depicted in Fig. 4C, the ultrafast TA spectra of BPP in acetonitrile solution ($c = 0.5$ mM) following 400 nm excitation (power intensity, 20 nJ per pulse) exhibit abundant excited state spectra features, including a ground state bleaching (GSB) at 420–430 nm, a simulated emission (SE) band at 430-500 nm, and two excited state absorption bands (ESA 1 and ESA 2) in the range of 500-650 nm (see Fig. S8 for details). The ESA 1 band at 540 nm shows single exponential decay of ~849 ps which

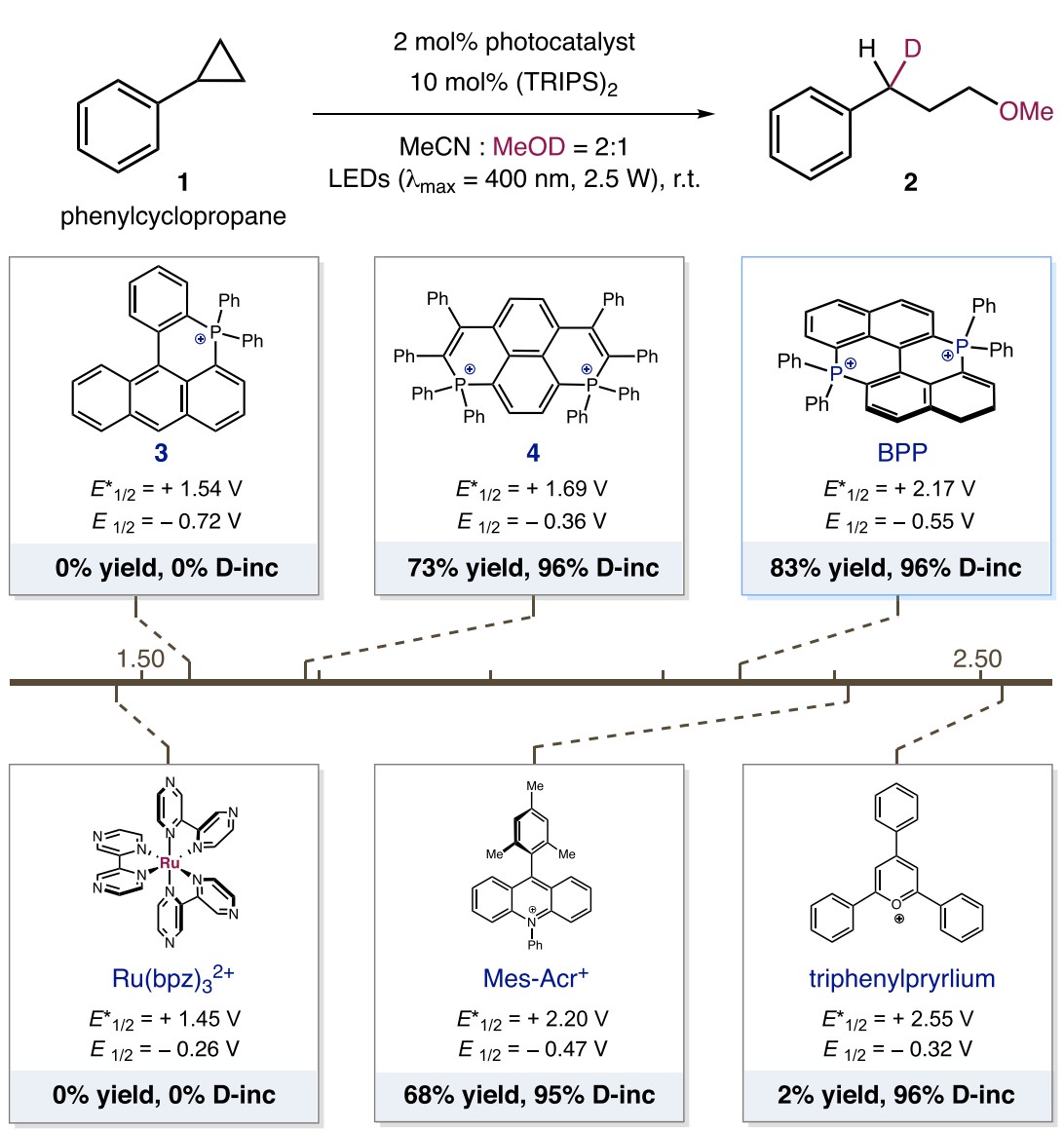

| entry | control experiments | yield[a] | D-inc[a] |
|---|---|---|---|
| 1 | no photocatalyst | 0% (92%) | n.d. |
| 2 | no (TRIPS)$_2$ | 0% (92%) | n.d. |
| 3 | no LED light | 0% (97%) | n.d. |
| 4 | with 10 equiv. H$_2$O | 82% (0%) | 22% |

**Fig. 2 | Reaction optimization and control experiments.** Reactions were performed in a parallel reactor, with 0.2 mmol **1**, 2 mol% photocatalyst, 10 mol% (TRIPS)$_2$, CH$_3$CN (0.6 ml) and MeOD (0.3 ml). [a]The yields were determined with GC-FID, the D-incorporation (D-inc) was determined by HR-MS. The recovery of **1** was presented in parentheses.

is accompanied by recovery of the SE band at 466-nm region indicating the decay dynamics of the ESA 1 and SE bands originate from the same singlet excited state (S1 state). The SE negative signal coincides with the fluorescence emission spectrum of BPP measured by fluorescence spectroscopy, and we identify it as the fluorescence signal generated when the S1 state returns to the ground state after BPP excitation. The lifetime ($\tau_f$ = 819 ps measured by TCSPC) is consistent with the decay lifetime ($\tau_1$ = 849 ps) obtained by transient spectral dynamics analysis.

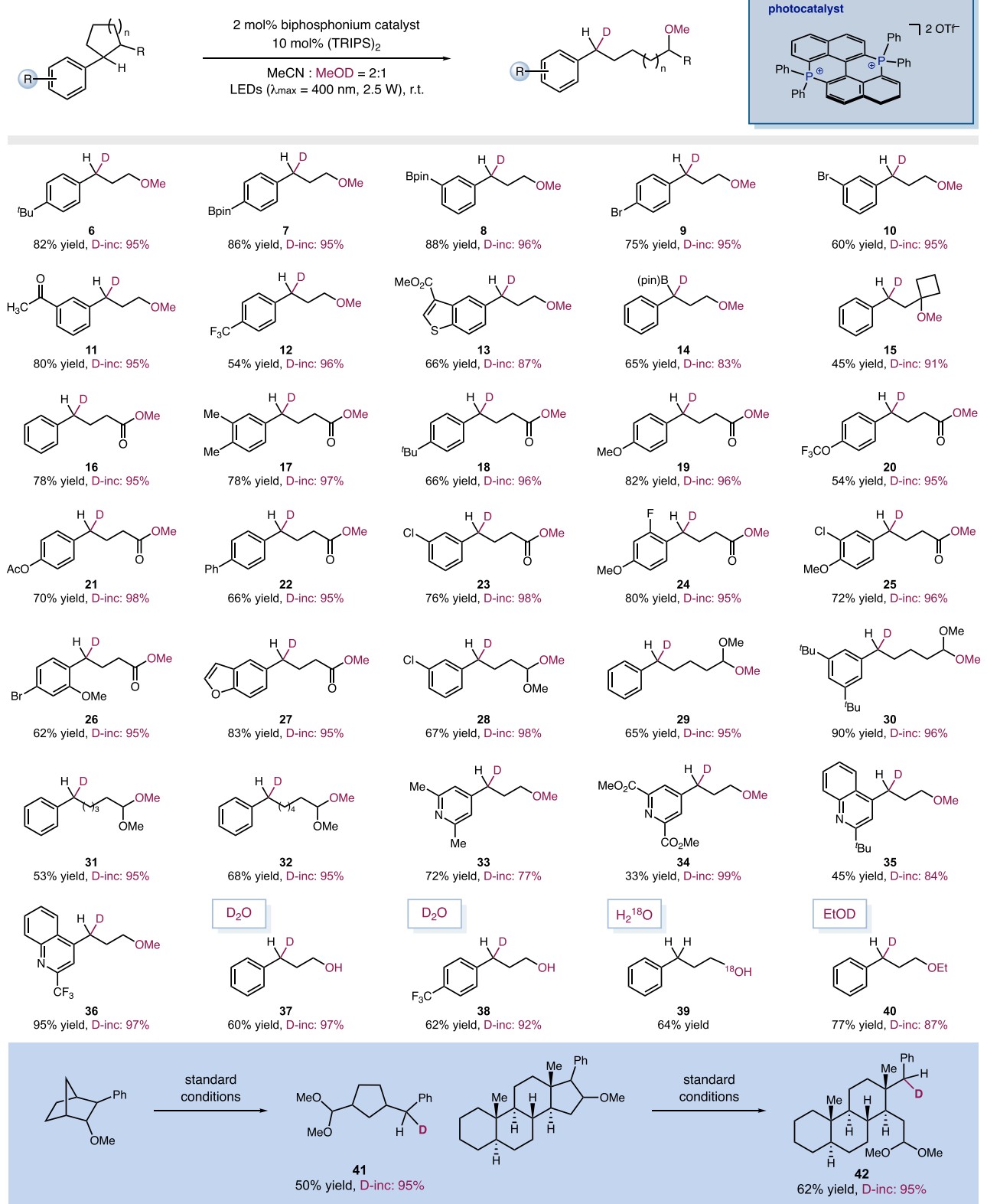

**Fig. 3 | Reaction scope.** Reactions were performed in a parallel reactor, with 0.2 mmol substrate, 2 mol% photocatalyst, 10 mol% (TRIPS)₂, CH₃CN (0.6 ml) and MeOD (0.3 ml). Deuterium incorporation was determined by HR-MS analysis.

The ESA 2 band at 625 nm showcases two exponential dynamics with a rise of 849 ps followed by an infinity-long lifetime decay. The rise component of 849 ps is reasonably ascribed to the intersystem crossing relaxation (ISC) time from the S1 state to the triplet state (T1)

state. The latter component corresponds to the decay lifetime of T1 state.

To obtain a more complete dynamic process of the ESA 2 band, we carried out transient absorption spectroscopy in ns-μs timescale in the

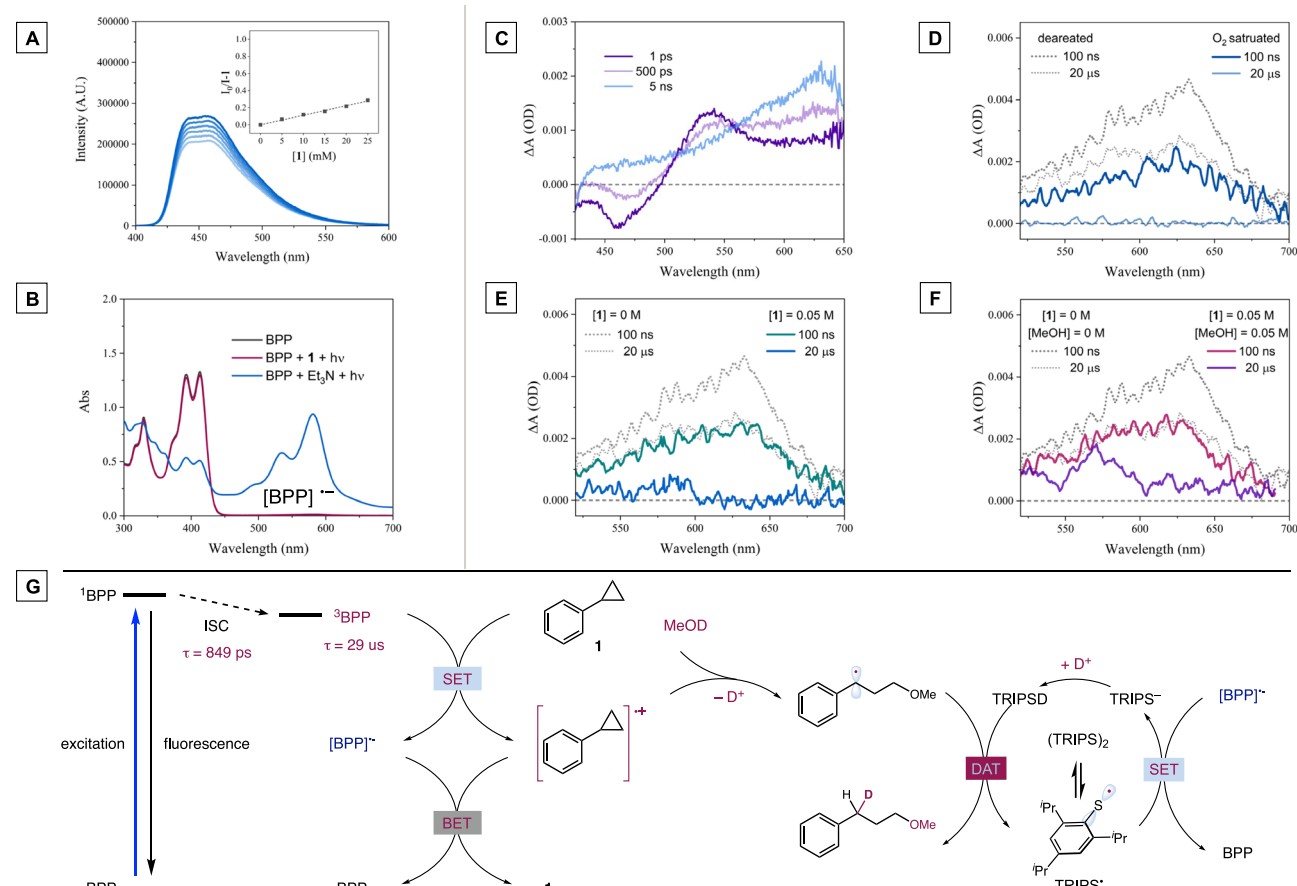

**Fig. 4 | Mechanistic investigations and proposed reaction pathways. A** Stern-Volmer quenching experiment. **B** Photolysis experiment. **C** Transient absorption spectra. **D** ESA decay signals in the presence of oxygen. **E** ESA decay signals in the presence of **1**. **F** ESA decay signals in the presence of **1** and MeOH. **G** proposed photoexcitation and monodeuteration pathways.

region of 525–700 nm. As shown in Fig. 4D, the ESA 2 band exhibits exponential decay with time constants of 29 μs. Monitoring whether the lifetime of the photoreaction system has a rapid reduction after the introduction of oxygen can be used to determine whether the triplet state exists. As demonstrated in Fig. 4D, oxygen can quickly quench 640 nm region signal, thus identifying the signal of ESA 2 band as the triplet signal generated by ISC after BPP excitation.

After the identification of the triplet state which is responsible for the desired electron transfer event, we next monitored the TA spectrum of BPP catalyst in the presence of phenylcyclopropane **1** (BPP: **1** = 1:100). With the addition of phenylcyclopropane, an accelerated decay of triplet BPP absorption can be observed (Fig. 4E). At 20 μs, the absorption centered of ESA 2 band at 625 nm returned to baseline, indicating a nearly complete consumption of triplet BPP by the SET with **1**. But in the 580 nm region, the characteristic absorption of [BPP]·⁻ was too weak to be distinguished. This peculiar observation could be explained by the fast and predominant back electron transfer between [BPP]·⁻ and radical cation of phenylcyclopropane [**1**·⁺] that has converted [BPP]·⁻ into ground state BPP. Thusly, no net electron transfer takes place and this gives rise to no [BPP]·⁻, in accordance with the observation made in the photolysis experiment. Nevertheless, in the presence of phenylcyclopropane and methanol, the characteristic absorption of [BPP]·⁻ can be clearly observed as the triple absorption returned to baseline (Fig. 4F). The nucleophilic attack of [**1**·⁺] with methanol generates of a more stable and less oxidizing benzylic radical through a ring-opening process. Thusly, the net formation of [BPP]·⁻ has indicated that the back-electron transfer event is suppressed by the nucleophilic attack of [**1**·⁺].

## Proposed reaction mechanism

Based on these findings, the photoinduced electron transfer processes in this ring-opening deuteration have been established and the proposed reaction pathway was depicted in Fig. 4G. Upon visible-light irradiation, a transient singlet BPP can be formed, most of which undergo a fast ISC process ($\tau_{ISC}$ = 849 ps) to generate a long-lived triplet state ($\tau_1$ = 29 us). The single electron oxidation of **1** by ³BPP is thermodynamically favorable and generates [**1**·⁺] and reduced BPP. Nevertheless, bimolecular back electron transfer from the resultant [**1**·⁺] to [BPP]·⁻ regenerates ground state catalyst, rendering a fully reversible PET process. The BET process could be suppressed by the well-established nucleophilic attack of [**1**·⁺] with methanol, forming a benzylic radical intermediate. A subsequent deuterium atom transfer mediated by TRIPSD would deliver the desired monodeuteration product. Lastly, a single electron transfer event would regenerate BPP and thiol catalysts.

In summary, we show a practical and selective monodeuteration protocol utilizing radical-mediated C–C bond scission and deuterium atom transfer processes. Under LED irradiation, in the presence of bisphosphonium photocatalyst, thiol catalyst, and CH₃OD, easily accessible cyclic hydroaromatics can be efficiently converted into mono-deuterated products equipped with distal functionalities. The redox-neutral and operationally simple reaction conditions provides a synthetically appealing approach for the constructions of benzylic CDH moieties. Through spectroscopy experiments including transient absorptions, the photoexcitation and photoinduced electron transfer events of bisphosphonium catalyst are established, elucidating a peculiar back-electron transfer process which can be regulated by subsequent nucleophilic additions.

## Methods

### General procedure for batch reactions

An 8 mL vial was charged with substrates (1.0 equiv.), BPP (0.02 equiv.), (TRIPS)$_2$ (0.1 equiv.) and 1 mL MeCN/MeOD ($v/v = 2{:}1$). The vial was sealed with a Teflon®-lined cap, the reaction mixture was degassed by argon sparging for 10 minutes. The mixture was then irradiated with LED ($\lambda_{max} = 400$ nm photon flux, 2.5 W) and stirred under irradiation at ambient temperature for 16 hours. After the reaction, the mixture was evaporated in vacuo and the residue was purified by flash chromatography.

## Data availability

Data are available in the manuscript and supplementary materials. Data supporting the findings of this manuscript are also available from the authors upon request.

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

## Acknowledgements

This work was supported by the National Key R&D Program of China (No. 2021YFA1500100), the National Natural Science Foundation of China (no. 22125111, 21971163), the Natural Science Foundation of Shanghai (21XD1424600), the Strategic Priority Research Program of the Chinese Academy of Sciences (XDB0610000), and the Shanghai Pilot Program for Basic Research—Chinese Academy of Science, Shanghai Branch.

## Author contributions

Y.X., W.C. contributed equally to this work. Z.Z. conceived and directed the research; Z.Z., Y.X., W.C., R.P., J.D., Q.A., W.L., and Y.Y.designed the experiments. Y.X., W.C., R.P., J.D., and Q.A. performed and analyzed the reactions; Z.Z., Y.X., and W.C. prepared the manuscript, which was approved by all authors.

## Competing interests

The authors declare no competing interests.
