## [Transparent Peer Review file · Nature Communications]

Selective monodeuteration enabled by bisphosphonium catalyzed ring opening processes

Corresponding Author: Professor Zhiwei Zuo

Version 0:

Reviewer comments:

Reviewer #1

(Remarks to the Author)

This work by Yang, Liu, Zuo, et al. describes a radical monodeuteration of phenyl cycloalkanes and cyclic ketones (ranging from 3 to 7 membered rings) using bisphosphonium (2 mol%) as the photocatalyst with violet LEDs and a disulfide (10 mol%) as the HAT catalyst in a 2:1 mixture of MeCN and MeOD at ambient temperature. The reaction shows a modest substrate scope in yields range from (50-90%), high deuterium incorporation (95-98%). The mechanistic study is well reasoned and supported by spectroscopy experiments to support the electron transfer events between the catalyst and substrate as well as the subsequent reactivity. I would argue that this work is interesting given this method may allow access monodeuterated derivatives, but it is not entirely surprising reactivity given the wealth of prior examples of ring-opening transformations have been developed over the last decade as noted by the authors in ref 37-49 as well as the strategy of radical monodeuterated via deuterated thiol catalyst HAT by Studer, Xie, Zhu, and others. In addition, the substrate scope is limited to basic phenyl substituted cyclo alkanes and ketones, is there any examples with N-heterocyclic ring system which may be of more valuable in a pharmaceutical setting? Is this methodology applicable to late-stage functionalization to demonstrate robustness? Have the authors considered the use of a chiral thiol or disulfide HAT catalyst to explore the possibility of an enantioselective deuteration process which would be of higher interest? I think this is a nice piece of work, but in its current form, I don't think it is novel enough for publication in Nat. Commun. I think should be published elsewhere, perhaps in a more specialized journal.

Other comments.

-Is there any diastereoselective control of the deuteration incorporation of product 31? What is the d.r.?

-Can the reaction be performed on 4 mmol or more with comparable level of yield, and deuterium incorporation to demonstrate synthetic utility and scalability. At larger scale could the amounts of reagents can be lowered (i.e MeOD, photocatalyst, and disulfide)?

-It would be helpful for the reader to include the structure (Figure 4G) of (TRIPS)₂ and what the abbreviation is in the manuscript (i.e bis(2,4,6-triisopropylphenyl) disulfide)

Reviewer #2

(Remarks to the Author)

In this manuscript, the authors have described photoredox-catalyzed ring-opening alkoxylation-deuteration reactions of aryl cycloalkanes. The reaction is a nice combination of photocatalytic aryl radical cation-mediated C–C bond cleavage and thiol-catalyzed deuterium atom transfer. Compared to previous reports on ring-opening bifunctionalization of aryl cyclopropanes via aryl radical cations, the major advance of this work is the expansion of the substrate structure from cyclopropanes to larger-sized rings. A range of deuterated ethers, esters, and acetals can be obtained in good efficiency with high deuterium incorporation. The photoexcitation and photoinduced electron transfer events of bisphosphonium catalyst have also been explored by spectroscopy experiments including transient absorptions. The work is solid and well crafted. Therefore, I recommend publication of this manuscript in Nat. Commun. after revisions.

Prior to publication, the following issues should be properly considered:

1. The nucleophilic reagents used are limited to methanol. The authors should explore the applicability of other alcohols and nucleophilic reagents.
2. Can the cycloalkane skeleton have another functional group or substituent?
3. Some relevant literature should be cited: Nat. Commun. 2019, 10, 4367; Chem. Eur. J. 2020, 26, 11690; Org. Lett. 2020, 22, 8681; Chem. Sci. 2022, 13, 12831; Angew. Chem. Int. Ed. 2023, 62, e202310671; J. Org. Chem. 2023, 88, 3787.

Reviewer #3

(Remarks to the Author)

In this manuscript, Zuo and co-workers describes a novel and facile photocatalytic selective monodeuteration protocol through bisphosphonium catalyzed ring opening processes. The novelty of this transformation is that it is a monodeuteration protocol with high selectivity, in which C–C bond as the unconventional functional handle to generation the radical intermediates. The synergistic combination of radical-mediated C–C bond scission and deuterium atom transfer processes has enabled the effective constructions of benzylic CDH moieties with good to excellent yields. This protocol offers numerous advantages, including mild reaction conditions, a wide range of suitable substrates, and excellent compatibility with functional groups. In particular, the fact that this work so well for the challenging less strained 5 to 7 ring size change and steroid ring systems (e.g. 26-31) shows that this part of the project is working very well, and the ring opening has been well designed. Finally, the investigation of mechanism in the paper was also well conducted, based on a series of mechanistic studies, including transient absorptions, the photoexcitation and photoinduced electron transfer events of bisphosphonium catalyst have been established, elucidating a peculiar back-electron transfer process which can be regulated by subsequent nucleophilic additions. The manuscript presents an interesting and valuable contribution to the field. Therefore, I suggest this manuscript to be published on Nature Commun.

Some minor corrections should be accomplished before publication:

1. Authors should include at least one sentence in the introduction section of the manuscript to illustrate the importance of photocatalysis.
2. Is it possible inexpensive and operationally handy deuteration agents D₂O could be employed in this transformation?
3. What is the relationship between different ring strain and reaction rate?
4. A related reference (ACS Catal. 2020, 10, 6603) might be better cited.

Version 1:

Reviewer comments:

Reviewer #1

(Remarks to the Author)

This reviewer appreciates the improvements the authors has made to the manuscript in addressing this reviewer comments/concerns. After further review, I believe in terms of novelty and of significance to the chemistry community this manuscript is acceptable to be published in Nat. Commun.

Reviewer #2

(Remarks to the Author)

In the revised manuscript, the authors have addressed all my concerns. Therefore, publication in Nature Communications is recommended.

Reviewer #3

(Remarks to the Author)

In their revised version, Zuo, Yang, Liu and co-workers answered the concern of the reviewers and have now an excellent work of high interest in the field of C–C bond cleavage and deuteration of benzyl C(sp³)-H bonds. The authors have carefully added some N-heterocyclic ring and drugs, continuous-flow synthesis approach and some new nucleophilic reagents, and answered all of the questions be the reviewers and adopted the suggestions. The synthetically important paper can now be published as it is.

Response to Reviewer 1:

This work by Yang, Liu, Zuo, et al. describes a radical monodeuteration of phenyl cycloalkanes and cyclic ketones (ranging from 3 to 7 membered rings) using biphosphonium (2 mol%) as the photocatalyst with violet LEDs and a disulfide (10 mol%) as the HAT catalyst in a 2:1 mixture of MeCN and MeOD at ambient temperature. The reaction shows a modest substrate scope in yields range from (50-90%), high deuterium incorporation (95-98%). The mechanistic study is well reasoned and supported by spectroscopy experiments to support the electron transfer events between the catalyst and substrate as well as the subsequent reactivity. I would argue that this work is interesting given this method may allow access monodeuterated derivatives, but it is not entirely surprising reactivity given the wealth of prior examples of ring-opening transformations have been developed over the last decade as noted by the authors in ref 37-49 as well as the strategy of radical monodeuterated via deuterated thiol catalyst HAT by Studer, Xie, Zhu, and others. In addition, the substrate scope is limited to basic phenyl substituted cycloalkanes and ketones, is there any examples with N-heterocyclic ring system which may be of more valuable in a pharmaceutical setting? Is this methodology applicable to late-stage functionalization to demonstrate robustness? Have the authors considered the use of a chiral thiol or disulfide HAT catalyst to explore the possibility of an enantioselective deuteration process which would be of higher interest? I think this is a nice piece of work, but in its current form, I don't think it is novel enough for publication in Nat. Commun. I think should be published elsewhere, perhaps in a more specialized journal.

We thank the reviewer for the comments. We want to further highlight the fact that, while the literature has documented ring-opening reactions and the application of the deuterated thiol-catalyst DAT, the concept of a metal-free, selective cleavage of unstrained C–C bonds, along with monodeuteration or other functionalizations, was previously uncharted territory. Our research pioneers this approach, and we consider our findings to be both novel and significant from a synthetic chemistry perspective. This assessment is corroborated by the positive feedback from peer reviewers.

1) is there any examples with N-heterocyclic ring system ... Is this methodology applicable to late-stage functionalization to...

We have expanded our investigation to include a comprehensive examination of the substrate scope, with a particular emphasis on heteroarenes. Notably, a range of heteroarenes, including N-heterocyclic ring, such as **pyridine** and **quinoline** derivatives, have been successfully accommodated, yielding

desired monodeuterated products with excellent to moderate yield, without any significant compromise to the D-incorporation efficiency.

Cyclopropane	Product	Yield (%) ^a	D-inc. (%) ^b
	 33	72	77
	 34	33	99
	 35	45	84
	 36	95	97
	 13	66	87

^a Determined by isolation. ^b Determined by HR-MS.

Regarding late-stage functionalization concern, substrate **31** has demonstrated the potential of our methodology as a versatile tool for the modification of complex molecular structures.

2) Have the authors considered the use of a chiral thiol or disulfide HAT catalyst...enantioselective deuteration process...

While integrating enantioselective processes into HAT approach holds significant promise for broadening its scope and potency, it is essential to acknowledge the complexities and formidable challenges. The nuanced differences between hydrogen and deuterium, despite their isotopic kinship, present considerable hurdles in the precise delineation and profiling of chiral centers (*J. Am. Chem. Soc.* **1994**, *116*, 9652; *Nature* **2007**, *446*, 526). Given the magnitude of these difficulties, and the fact that methylene centers were produced in the current method, we currently did not test chiral thiols; but for subsequent projects concerning tertiary carbon centers where asymmetric

deuteration would be more useful, we will continue to explore chiral HAT catalysts.

3) Is there any diastereoselective control of the deuteration incorporation of product 31? What is the d.r.?

At the current stage, we encounter a limitation in our capacity to discern between the isomers of compound 31. Only one set of proton can be observed in ^1H NMR, and our current analytical capabilities have yet to surmount the subtleties that differentiate these isomeric forms.

4) Can the reaction be performed on 4 mmol or more with comparable level of yield, and deuterium incorporation to demonstrate synthetic utility and scalability. At larger scale could the amounts of reagents can be lowered (i.e MeOD, photocatalyst, and disulfide)?

We have successfully implemented a continuous-flow synthesis approach, enabling scaled-up production capabilities. Under the optimized reaction conditions, we achieved an 87% yield along with a space-time yield (STY) of 17.4 g/L·h. Utilizing flow conditions significantly reduces the amount of the disulfide loading to 5 mol%. Moreover, lower quantities of MeOD could be used to achieve satisfactory levels of efficiency.

Comprehensive experimental details are provided in the ESI (page S60–S61).

5) It would be helpful for the reader to include the structure (Figure 4G) of (TRIPS)₂ and what the abbreviation is in the manuscript (i.e bis(2,4,6-triisopropylphenyl) disulfide)

All changes were made as per request.

Response to Reviewer 2:

In this manuscript, the authors have described photoredox-catalyzed ring-opening alkoxylation-deuteration reactions of aryl cycloalkanes. The reaction is a nice combination of photocatalytic aryl radical cation-mediated C-C bond cleavage and thiol-catalyzed deuterium atom transfer. Compared to previous reports on ring-opening bifunctionalization of aryl cyclopropanes via anyradical cations, the major advance of this work is the expansion of the substrate structure from cyclopropanes to larger-sized rings. A range of deuterated ethers, esters, and acetals can be obtained in good efficiency with high deuterium incorporation. The photoexcitation and photoinduced electron transfer events of bisphosphonium catalyst have also been explored by spectroscopy experiments including transient absorptions. The work is solid and well crafted. Therefore, I recommend publication of this manuscript in Nat. Commun. after revisions.

We thank the reviewer for the comments.

1) The nucleophilic reagents used are limited to methanol. The authors should explore the applicability of other alcohols and nucleophilic reagents.

In our additional experimental series, we employed a range of deuterated nucleophiles and isolated the targeted monodeuterated products. However, concomitant with this synthesis, we registered a gradual attenuation in the level of deuterium incorporation. The nucleophiles evaluated included D₂O, EtOD, 1-heptanol-*d*, 2-phenylethan-1-ol-*d*, 2,4,6-triisopropylbenzenethiol-*d*, and a deuterated sugar derivative 1,2:3,4-di-O-isopropylidene-D-galactopyranose-*d*. This observed decrease in D-inc. seems to correspond with a reduction in the reaction rate between the thiolate ions and deuterium. Although we hypothesize that this reaction deceleration might be influenced by factors such as the pK_a values and steric hindrance of the nucleophiles, we emphasize that these are preliminary observations. Moreover, ¹⁸O labeling can be easily achieved using H₂¹⁸O as the nucleophile under standard conditions. The experimental details were documented in the ESI (page S48–49, S62–S64).

NuD	Product	Yield (%) ^a	D-inc. (%) ^a
D ₂ O	 37	60	97
EtOD	 40	77	87
ⁿ HeptOD	 R1	57	73
Ph-CH ₂ -CH ₂ -OD	 R2	70	71
TRIPSD	 R3	33	52
galactose-d ^b	 R4	33	38
H ₂ ¹⁸ O	 39	64	-

^a Determined by HR-MS. ^b 0.1 mmol scale and 5.0 eq. of nucleophile was used.

2) Can the cycloalkane skeleton have another functional group or substituent?

In our experimental endeavors, we conducted reactions utilizing 4,4,5,5-tetramethyl-2-(1-phenylcyclopropyl)-1,3,2-dioxaborolane and 1-phenylspiro[2.3]hexane as the starting materials. Through our methodology, we have successfully achieved the formation of the targeted monodeuterated derivatives, which were procured in moderate yields alongside good levels of D-inc. (**14** and **15**).

In an effort to diversify the molecular scaffolding, we tested eight different phenyl cyclic ethers, lactones, and their derivatives under standard conditions.

Examples of these compounds include 2-phenyloxetane, 3-methoxy-4-phenyltetrahydrofuran, and 3-phenyloxetan-2-one. However, it was observed that the starting materials did not undergo transformation in these experiments.

The experimental details were documented in the ESI (page S35–S36, S51, Scheme S6).

3) *Some relevant literature should be cited: Nat. Commun.* **2019**, *10*, 4367; *Chem. Eur. J.* **2020**, *26*, 11690; *Org. Lett.* **2020**, *22*, 8681; *Chem.Sci.* **2022**, *13*, 12831; *Angew. Chem. Int. Ed.* **2023**, *62*, e202310671; *J. Org. Chem.* **2023**, *88*, 3787.

All changes were made as per request.

Response to Reviewer 3:

In this manuscript, Zuo and co-workers describes a novel and facile photocatalytic selective monodeuteration protocol through bisphosphonium catalyzed ring opening processes. The novelty of this transformation is that it is a monodeuteration protocol with high selectivity, in which C–C bond as the unconventional functional handle to generation the radical intermediates. The synergistic combination of radical-mediated C–C bond scission and deuterium atom transfer processes has enabled the effective constructions of benzylic CDH moieties with good to excellent yields. This protocol offers numerous advantages, including mild reaction conditions, a wide range of suitable substrates, and excellent compatibility with functional groups. In particular, the fact that this work so well for the challenging less strained 5 to 7 ring size change and steroid ring systems (e.g. 26-31) shows that this part of the project is working very well, and the ring opening has been well designed. Finally, the investigation of mechanism in the paper was also well conducted, based on a series of mechanistic studies, including transient absorptions, the photoexcitation and photoinduced electron transfer events of bisphosphonium catalyst have been established, elucidating a peculiar back-electron transfer process which can be regulated by subsequent nucleophilic additions. The manuscript presents an interesting and valuable contribution to the field. Therefore, I suggest this manuscript to be published on Nature Commun.

We thank the reviewer for the comments.

1) Authors should include at least one sentence in the introduction section of the manuscript to illustrate the importance of photocatalysis.

As suggested, the following sentence has been added in the introduction section: “emerging photoredox catalysis as the enabling platform to generate open-shell radical intermediates, have provided a synthetically valuable strategy to exploit ubiquitous C(sp³)–C(sp³) bonds as functional handles”.

2) Is it possible inexpensive and operationally handy deuteration agents D₂O could be employed in this transformation?

D₂O can be employed as the deuteration agent under the standard conditions to furnish good levels of results (product 37 and 38 in the updated Fig. 3) Moreover, ¹⁸O labeling can be easily achieved using H₂¹⁸O as the nucleophile under standard conditions.

3) What is the relationship between different ring strain and reaction rate?

In our kinetic analysis, a structured series of oxygen-substituted cycloalkane derivatives, specifically (2-methoxycyclobutyl)benzene, (2-methoxycyclopentyl)benzene, (2-methoxycyclohexyl)benzene, and 1-methoxy-2-phenylcycloheptane, were employed as substrates. This systematic selection aimed to delineate the interplay between ring strain and reaction kinetics. Our experimental outcomes demonstrate a positive correlation, signifying that an increase in ring strain is associated with an accelerated reaction rate.

The experimental details were documented in the ESI (page S19, Figure S17).

4) A related reference (*ACS Catal.* 2020, 10, 6603) might be better cited.

All changes were made as per request.